# Current Challenges in Image-Guided Magnetic Hyperthermia Therapy for Liver Cancer

**DOI:** 10.3390/nano12162768

**Published:** 2022-08-12

**Authors:** Anirudh Sharma, Erik Cressman, Anilchandra Attaluri, Dara L. Kraitchman, Robert Ivkov

**Affiliations:** 1Department of Radiation Oncology and Molecular Radiation Sciences, The Johns Hopkins University School of Medicine, Baltimore, MD 21231, USA; 2Department of Interventional Radiology, Division of Diagnostic Imaging, MD Anderson Cancer Center, Houston, TX 77030, USA; 3Department of Mechanical Engineering, School of Science, Engineering, and Technology, The Pennsylvania State University, Middletown, PA 17057, USA; 4Department of Radiology and Radiological Science, Johns Hopkins University School of Medicine, Baltimore, MD 21205, USA

**Keywords:** magnetic nanoparticle, hyperthermia, hepatocellular carcinoma, perfusion imaging, specific loss power, temperature feedback control

## Abstract

For patients diagnosed with advanced and unresectable hepatocellular carcinoma (HCC), liver transplantation remains the best option to extend life. Challenges with organ supply often preclude liver transplantation, making palliative non-surgical options the default front-line treatments for many patients. Even with imaging guidance, success following treatment remains inconsistent and below expectations, so new approaches are needed. Imaging-guided thermal therapy interventions have emerged as attractive procedures that offer individualized tumor targeting with the potential for the selective targeting of tumor nodules without impairing liver function. Furthermore, imaging-guided thermal therapy with added standard-of-care chemotherapies targeted to the liver tumor can directly reduce the overall dose and limit toxicities commonly seen with systemic administration. Effectiveness of non-ablative thermal therapy (hyperthermia) depends on the achieved thermal dose, defined as time-at-temperature, and leads to molecular dysfunction, cellular disruption, and eventual tissue destruction with vascular collapse. Hyperthermia therapy requires controlled heat transfer to the target either by in situ generation of the energy or its on-target conversion from an external radiative source. Magnetic hyperthermia (MHT) is a nanotechnology-based thermal therapy that exploits energy dissipation (heat) from the forced magnetic hysteresis of a magnetic colloid. MHT with magnetic nanoparticles (MNPs) and alternating magnetic fields (AMFs) requires the targeted deposition of MNPs into the tumor, followed by exposure of the region to an AMF. Emerging modalities such as magnetic particle imaging (MPI) offer additional prospects to develop fully integrated (*theranostic*) systems that are capable of providing diagnostic imaging, treatment planning, therapy execution, and post-treatment follow-up on a single platform. In this review, we focus on recent advances in image-guided MHT applications specific to liver cancer

## 1. Introduction

Despite considerable progress that includes immunotherapies, liver transplantation remains the best option to extend life with improved quality for patients diagnosed with advanced (unresectable) hepatocellular carcinoma (HCC). Considering the enormous and growing gap between organ supply and demand, palliative non-surgical options have become the default front-line treatments for many patients [1,2,3,4,5,6]. Options include chemotherapy, radiation, ablation, transarterial chemoembolization (TACE), and radioembolization, but each of these comes with its own unique risks and benefits. Challenges to patient safety, treatment delivery or implementation, and skill-dependent treatment effectiveness vary substantially. Perhaps as a result, the successes remain generally inconsistent and below expectations, and with the exception of radioembolization, three-year survival rates are <10% [6,7]. New approaches, or perhaps novel combinations of existing approaches, are needed to raise the prospects for patients diagnosed with advanced and unresectable HCC.

### 1.1. Imaging-Guided Interventional Approaches Offer Benefits

Among the treatment options available to cancer patients, imaging-guided interventions have emerged as attractive procedures that offer individualized tumor targeting. These interventions include methods where tumor localization and drug delivery are guided and monitored by noninvasive imaging. For HCC, often being multifocal and advanced at presentation, a logical and attractive method of treatment is non-invasive imaging-guided intervention that aids in the selective targeting of tumor nodules without impairing liver function. Furthermore, imaging-guided treatments can target standard-of-care chemotherapies to the liver tumor directly to reduce the overall dose and limit toxicities commonly seen with systemic administration of the same agents. In cases where the tumor has metastasized to other parts of the body, palliation by this approach often becomes the only practical option to modestly extend survival. 

Recent innovation and clinical translation of both magnetic and optical near-infrared (NIR) tracers has provided new tools for cancer diagnosis, image-guided treatment planning, and intra-operative guidance [8,9,10]. Magnet-based imaging modalities (e.g., MRI and MPI) offer the advantage of non-invasive, deep-tissue sensing from a distance. “Line-of-sight” is not required because tissue is diamagnetic and does not appreciably attenuate magnetization, which provides the signal, produced by magnetic nanoparticles localized in tissue(s). In contrast, NIR-tracers require detection in close proximity (<1 cm distances) because tissue attenuates IR/NIR light via absorption, primarily by water molecules [11]. This tissue attenuation requires the insertion of (usually fiber-optic based) imaging devices into the patient to identify deep-seated tumors/nodules. 

Quantitative imaging is possible with magnetic tracers (Section 2.5)—a feature integral to developing accurate MHT treatment plans (Section 2.13). Conversely, quantitation with NIR-imaging dyes is complicated by potential quenching reactions in varied biological environments, and the inherently variable optical properties of tissue (e.g., bone). NIR-imaging tracers, however, can provide intraoperative guidance during surgical procedures [8], which can complement the use of magnetic particle imaging-based diagnosis and pre-surgical planning, especially in cases where surgical resection of the tumor is warranted. Additionally, recent advances in chemistry have produced NIR dyes linked to small molecule (<40 kDa) targeting ligands [8,12], enabling rapid intra-operative diagnosis resulting from faster pharmacokinetics, which in turn complements quantitative imaging with magnetic tracers. 

### 1.2. The Promise of Thermal Medicine and Challenges to Its Implementation

Heat is mechanical incoherent (non-ionizing) energy that has profound effects on biology and living organisms. Exposure to elevated temperature increases the cellular and physiologic stress that, depending on temperature and duration of exposure, leads to molecular dysfunction, cellular disruption, and eventual tissue destruction with vascular collapse. Given its profound effects on cancer, which have been recognized for over 2000 years, hyperthermia in cancer treatment remains surprisingly limited. This may be due to the technological challenges encountered with controlling the energy delivery to meet the current standards of precision medicine, and to the complexities of integrating heat-based procedures into a modern clinical workflow. Thermal medicine requires controlled heat transfer to the target either by in situ generation of the energy or its on-target conversion from an external radiative source. Hyperthermia is a non-ablative heat therapy that aims to achieve and maintain a target temperature between 39 °C to 47 °C. Data obtained from experiments with preclinical models and from human clinical trials demonstrate that overall disease responses improve substantially when *quality* hyperthermia is combined with other non-surgical standard-of-care therapies such as radiation, chemotherapy, or immune checkpoint inhibitors [13,14]. Treatment quality can be quantified by comparing the achieved thermal dose (time at temperature) with that prescribed in the treatment plan [15,16]. Thus, to achieve quality hyperthermia, both controlled energy delivery to the target and a complete temperature history of the treatment site are required. These requirements present technological challenges that have only been recently overcome by developments in energy delivery and imaging-based (e.g., magnetic resonance) thermometry. 

### 1.3. Magnetic Hyperthermia Offers Unique Solutions for Thermal Medicine

Magnetic hyperthermia (MHT) is a nanotechnology-based thermal therapy that exploits energy dissipation (heat) from forced magnetic hysteresis of a magnetic colloid. Magnetic hysteresis, or the lag between magnetization (output) of a material exposed to an external magnetic field (input), manifests in fluidic suspensions of *some* magnetic colloids as a *rate dependent,* non-linear response to an external alternating magnetic field (AMF) [17,18,19]. A key advantage offered by MHT over other HT modalities is that therapeutic heat deposition occurs within the tumor, offering potential for patient-specific precision therapy [19,20,21,22,23]. MHT with magnetic nanoparticles (MNPs) and alternating magnetic fields (AMFs) requires the targeted deposition of MNPs into the tumor, followed by exposure of the region to an AMF [15,24,25] (Figure 1). In contrast to convective hyperthermia methods, where the energy delivered to the tumor is limited by heat transfer, heating tumors internally using MNPs allows for the scaling of MHT to the clinic. The European Medicines Agency approved MHT in 2010 for use in humans to treat recurrent glioblastoma in combination with radiation therapy [21,22]. MNPs offer additional functionality for imaging. This capability benefits MHT by enabling prospective treatment planning, which can mitigate risk when volumetric thermometry is impractical or unavailable [26,27,28,29,30].

MNP-based MHT offers (i) controlled heating; (ii) minimal invasiveness of (remote-controlled) hyperthermia; (iii) integrated imaging; and (iv) scalability. Emerging modalities such as magnetic particle imaging (MPI) offer additional prospects to develop fully integrated (*theranostic*) systems that are capable of providing diagnostic imaging, treatment planning, therapy execution, and post-treatment follow-up on a single platform [31,32]. In this review, we focused on recent advances in image-guided MHT applications specific to liver cancer.

## 2. Current Challenges with MHT

### 2.1. MHT Requires Imaging of MNP Concentration and Distribution in Tissues

X-ray computed tomography (CT), magnetic resonance imaging (MRI), and positron emission tomography (PET) are imaging modalities that have aided in the evolution of modern radiation therapy (RT). At least one of these is integrated with all modern RT planning workflows. In similar fashion, MHT requires MNP-specific imaging technology that can characterize MNP concentration and distribution in tissues to enable robust treatment planning and post-treatment evaluation for quality assurance. Given MNPs are the heat source, the effectiveness of MHT strongly depends on their distribution in tissue [20,29,30]. Volumetric power deposition within the tumor will depend on the local concentration of MNPs. Any MNP-specific imaging capability will require the high spatial and contrast resolution of MNPs with the clear distinction of adjacent anatomical structures. The magnetic properties of MNPs and their inherent responsiveness to magnetic fields provides the capability for both imaging and heating, thus offering a natural platform for imaging-guided, patient-specific HT. 

### 2.2. MRI of MNPs for MHT Has a Narrow Quantifiable Range

Various MNP formulations have been used as clinical contrast agents for MRI since the 1980s [33,34,35]. MRI generates images from anatomical variations of time-dependent relaxations of (water-based) proton nuclear magnetic moments. When placed into the large magnetic field of a clinical MRI scanner, some MNPs manifest a large magnetization that exerts a powerful magnetic damping field on the surrounding proton moments. This effect produces regions of hypointense, or “negative” contrast, in the resulting images. Not surprisingly, the primary clinical use of MNPs has been as negative contrast agents for liver and lymph node MRI, with field-dependent relaxivities exceeding 100 mM^−1^s^−1^ [34,35,36,37].

In the context of treatment planning for MHT, the negative signal or T2*-contrast produced by MNPs in MRI limits their utility for spatial localization or iron quantification [35]. This limitation is further extended by the requirement that 50–100 mg Fe of MNP/g tissue is needed for effective MHT, exceeding the MRI limits. Tissues containing high concentrations of MNPs sufficient for MHT often generate strong artifacts in MR images, rendering them ineffective for further medical use including treatment planning. In addition, hypointense imaging features generated by MNP contrast agents within acceptable limits, cannot be distinguished from naturally occurring intensity variations in tissues that include endogenous sources such as hemorrhagic (high iron) deposits and air–tissue interfaces (e.g., lungs, skin surface, bowels, etc.). 

Efforts to develop MR sequences that generate “positive contrast” from MNPs have met with some success [33,38]. Stuber et al. developed an MRI methodology called inversion recovery with ON-resonant water suppression (IRON) to generate a hyperintense positive contrast signal from iron oxide nanoparticle-labelled stem cells while attenuating the signal from fat tissue and the background. They showed that the volume of the positive signal increased linearly with the number of labelled cells [39]. Zhang et al. demonstrated potential with an echoless pulse sequence, sweep imaging with Fourier transformation (SWIFT), to quantify iron in liver, spleen, and kidneys using T1 contrast [38]. They showed a linear correlation between tissue iron concentration and relaxivity, R1, and between ex vivo tissue heating and R1 for iron concentrations up to 3.2 mg Fe/g tissue. While this is a significant step to quantify iron concentration in organs when MNPs are systemically delivered, percutaneous injection of MNPs into the tumor deposits much higher local iron concentrations, thus exceeding the limits of quantification even by this method. In a liver HCC model, SWIFT imaging may enable the quantification of relative iron concentrations in the tumor vs. average iron concentration in normal liver parenchyma following delivery through hepatic artery. However, MNP concentrations >5 mg Fe/g tissue appear as a saturated local intensity in R_1_ maps and are potentially confounded by dipole artifacts. Most importantly, the quantification of local tissue concentrations of MNPs at such high iron concentrations using MRI is unreliable because they are indirectly detected through their effects on proton relaxation. Thus, an imaging modality able to directly measure signal from MNPs is needed.

### 2.3. Imaging Guidance for TACE and ThermoTACE Indications in Unresectable HCC

For intermediate stage HCC, transcatheter arterial chemoembolization (TACE) is the standard of care, based on the most widely used Barcelona Clinical Liver Cancer classification (BCLC) [40,41]. The intervention takes advantage of the fact that hypervascular primary and metastatic liver tumors are supplied by the hepatic artery, as opposed to the portal vein supplying the bulk of the liver. This differential blood supply to the tumor allows for targeting and ischemia of the tumor nodules through intra-arterial delivery of chemotherapeutic and embolic agents (e.g., drug-eluting embolics), with minimal damage to surrounding liver parenchyma. Furthermore, combining TACE with thermal therapies (thermoTACE) has clinically shown significant improvements measured by odds and risk ratios vs. monotherapy [42]. However, the overall response of HCC to TACE (and thermoTACE) is governed by multiple factors including patient-specific (albumin-bilirubin (ALBI) score, liver function, comorbidities), treatment-specific (favorable anatomy, allergic reactions), and tumor-specific (tumor size, boundaries, number of lesions, hypervascularity) factors [43]. Not all intermediate stage HCC patients responded favorably to TACE. For example, tumors larger than 5 cm in diameter responded poorly and are associated with post-embolization syndrome, which includes symptoms of nausea, pain in the right upper quadrant, vomiting, and fever in TACE-treated patients [43], and requires acute management. As another example, direct percutaneous injection within the tumor, if accessible, may be more favorable than TACE in cases where excessive parasitic arterial tumor-feeders and shunting, identified by performing pre-treatment scans (e.g., 99mTc), might preclude selective targeting [43]. Intra-arterial delivery of MNPs for MHT might need additional screening to rule out factors such as leakage through arterio-venous shunts, iron dosage limits, and unintended occlusion of capillaries from MNP aggregates. Additionally, MHT would be ineffective for treating very small (microsatellite) tumor nodules (<1 mm) [44]. Therefore, further patient stratification for suitability to TACE and thermoTACE, within the intermediate stage HCC group, based on the imaging of tumor physiology, burden, and assessment of liver function is recommended [43].

### 2.4. MNPs + X-ray Contrast Fluids Are Feasible Dual-Contrast Agents for HCC Imaging

Lipiodol^®^ (ethiodized oil) is an oil-based radiopaque contrast agent containing iodine at 480 mg/mL, which is indicated for several uses including the imaging of HCC via hepatic intra-arterial delivery [45]. Though its composition is unknown, it has been reported to be selectively taken up by tumors, with microembolic, drug carrying, and tumoricidal effects [5,46]. Lipiodol has been extensively used to image and treat liver tumors as part of the imaging-guided interventions such as transcatheter arterial chemoembolization (TACE), following pharmacy compounding of individualized chemotherapeutic cocktails. 

Attaluri et al. demonstrated feasibility to co-formulate MNPs (Bionized Nanoferrite^®^, or BNF nanoparticles) with lipiodol for dual-contrast CT/MRI imaging, and heating [46]. They showed that BNF-lipiodol (BNF-Lip) generated a higher measured increase in temperature (normalized by iron mass) compared to aqueous BNF formulations, in vitro and in vivo in xenograft HepG2 tumors in mice. The higher rise in temperature implied higher measured heating rates, but, with further measurements, both were attributed to the lower specific heat of lipiodol compared to that of water, despite a slightly decreased specific loss power estimate arising from altered crystallite arrangements in BNF cores [47]. Following X-ray-guided intra-arterial delivery of MNP-lipiodol, MR (T1) and CT were performed 7 days post-injection and co-localization of MNPs with lipiodol in the tumor region was qualitatively observed. However, the analysis of the data suggests that MNPs and lipiodol are prone to separate once delivered to tissue. Thus, post-injection separation of carrier fluid from MNP must be considered when planning MHT treatments using this combination. Additionally, the quantification of MNP concentration in tissues was confounded by susceptibility artifacts in MR imaging, and saturation at high MNP concentrations, thus precluding its use for quantitative computational modeling in treatment planning. Therefore, to accurately predict the heat generation and its transfer to, and throughout, the tumor, an MNP-specific imaging modality will be required. 

### 2.5. Magnetic Particle Imaging Offers MNP-Tracer Specific Imaging of Distribution and Concentration in Tissue

Magnetic particle imaging (MPI) is a highly sensitive MNP tracer-specific imaging method that relies entirely on the magnetization and magnetic properties of the MNPs to generate an image [31,32,48]. Akin to nuclear medicine, the imaging contrast and resolution are entirely tracer-dependent and are not attenuated by tissue. Thus, MPI allows for the linear quantitation of MNP concentration and highly specific imaging of regions containing MNPs regardless of anatomy. Co-registration of MPI with MRI or CT images is needed to ascertain the spatial distribution of MNPs relative to anatomical structures. MPI scanners use a combination of NdFeB permanent, or electro-magnets to create a static gradient magnetic field to generate a low-field, sensitive region (i.e., field free region or FFR) at the isocenter of the MPI magnet. Within the FFR, MNPs are “free” to respond to AC magnetic fields generated by an RF drive coil, and therefore produce a time-varying signal, derived from their collective field dependent magnetization response. Outside the FFR, the magnetic moments of the MNPs are “locked” in alignment with the direction of the DC field, and therefore, do not produce any signal in response to activation from the RF drive field. The FFR is rastered across the sample by moving the sample stage and/or the magnets, to generate images of the MNPs [49]. In contrast to MRI, MPI allows for imaging in hemorrhagic tissue and at the air–tissue interfaces. Clinical MPI is not yet available, but human functional MPI scanners are under development [50]. 

### 2.6. Liver Perfusion Imaging Can Provide Non-Invasive Diagnostic Imaging Modality

A normal human liver receives approximately 75% of its blood supply from the portal vein. The remaining 25% is provided by the hepatic artery [51]. Changes to liver perfusion signal potential global and regional alterations in the balance of arterial and venous blood flow(s) that can signal disease. Indeed, many end stage liver diseases including HCC and cirrhosis are characterized by relatively increased arterial flow with correspondingly reduced venous flow. Characterizing such regional and global changes in arterial and venous perfusion by using imaging methods offers potential avenues for highly sensitive and specific non-invasive diagnostic imaging.

In normal liver, blood is sourced from the portal vein and the hepatic artery, combining to form the portal triad, together with the bile ducts (Figure 2A,B). Blood flows from the portal triad through the sinusoids, which is flanked by rows of hepatocytes and lined by highly fenestrated sinusoidal endothelial cells (LSECs), through the space between the hepatocytes and LSECs (space of Disse) and, finally drains through the central vein. The collection of the portal triad, sinusoids, hepatocytes, and central vein, forms a repetitive unit throughout the liver, called the acini [52,53,54,55,56] (Figure 2). 

The resolution of diseased vs. healthy tissues is possible using perfusion imaging by quantifying the perfusion parameters that describe the flow, resistance, and contrast kinetics measured in the portal vein, heptatic artery, central vein, and the whole liver. These parameters are linked intimately with the liver architecture [57,58]. Secondary quantitative expressions (e.g., perfusion index = A/(A + P)) comparing the diseased vs. normal livers can be derived from these basic parameters [59,60]. In cirrhotic livers and HCC, architectural changes in the acini such as increased collagen deposits in the space of Disse, increased capillary density, and angiogenesis of tumor-associated, non-triadal branches (not associated with the portal triad) and compensatory portal venous-systemic shunts are common. These phenomena are typically characterized by high portal venous resistance (lower flow), increased arterial flow due to neovascularization, increased transit time, increase in arterial flow to compensate for lower portal venous flow (hepatic arterial buffer response), and overall higher serum VEGF [61,62,63]. 

To be useful, the perfusion imaging method must resolve changes in perfusion parameters from normal liver parenchyma vs. low-grade dysplastic nodules vs. HCC. This is generally possible if [60]:Spatial and kinetic differences of perfusion/flow between small nodules and HCC can be accurately resolved.Arterial and venous flow can be accurately quantified.Image-tracer concentration can be accurately differentiated from the contrast.

To establish perfusion imaging for diagnosing diseased livers, data providing the metrics listed above must be coupled with validated fitting and modeling methods to extract relevant perfusion parameters. Post-treatment efficacy is partly evaluated based on the contrast enhancement characteristics and recovery of the perfusion parameters (and secondary derived parameters) to the baseline healthy liver values.

### 2.7. MHT Treatment Planning Models Require Experimental Validation

Computational modeling enables the analysis of a variety of biophysical processes relatively economically, in order to improve the understanding of the underlying physiological processes for a range of system parameters and properties in the normal vs. diseased state, and thus aid in diagnosis and treatment. Computational models for heat transfer in hyperthermia treatment can provide a predictive tissue thermal dose for various heat inputs to facilitate image-guided therapy, but these models require the relevant tissue parameters as inputs and need to be validated experimentally. For MHT, computational modeling allows the investigator to evaluate the influence of specific variables on temperature distributions within tissues, provided that the heat sources (i.e., MNP concentration and distribution) and heat “sink” terms (i.e., perfusion) are known [26]. Treatment planning models in MHT are currently mostly based on Penne’s bioheat transfer equation where heat deposited from MNPs (through hysteresis) and eddy currents (through tissue-AMF Joule heating) is defined as the heat source terms, and heat loss through conduction, convection, and temperature-dependent blood perfusion in the tumor and surrounding tissue is defined as the heat sink terms [29,30,64].
(1)$$\rho_{n}c_{n}\frac{\partial T_{n}}{\partial t}=k_{n}{𝛻}^{2}T_{n}+\rho_{b}c_{b}\omega_{b}\left(T_{b}-T_{n}\right)+Q_{m,n}+Q_{eddy}+Q_{p}$$
where subscripts *n* and *b* represent the tissue/tumor and blood, respectively. $\rho_{n},\;c_{n},\;k_{n},\;T_{n},\;\;\mathrm{and}\;Q_{m,n}$ are the tissue/tumor density, specific heat, thermal conductivity, local temperature, and metabolic heat generation, respectively. t is the heating time; *Q_eddy_* is the heating rate per unit volume due to eddy currents; and *Q_p_* is the heating rate per unit volume of tumor due to nanoparticles. $\rho_{b},\;c_{b},\;\omega_{b}$ and $T_{b}$ denote the density, specific heat, perfusion rate, and temperature of blood, respectively.

As a core component of treatment planning, predictive modeling allows for the in silico assessment of a treatment over a range of simulated experimental conditions. Successful modeling enables the efficient use of time and resources by enabling a focus on treatment conditions that are likely to be successful, and identifying treatment failure modes and safety considerations. However, without appropriate verification and experimental validation, the reliability of computational models is undetermined. Verification, the assessment of accuracy with known solutions (often analytical), and validation, the assessment of the accuracy by comparison with experimental data, are essential to establish the accuracy and reliability of the computational models as appropriate medical tools [65]. Kandala et al. reported the validation and verification of a coupled electromagnetic and heat transfer model during MHT using agarose gel phantoms [66]. The computed temperatures in the gel phantom were in agreement with the analytical calculations of the eddy current heating (error < 1%) and temperatures measured experimentally (absolute error < 2%). Once validated, the coupled electromagnetic and heat transfer model was used to simulate treatments in a 3D rabbit liver HCC model. In this model, constant perfusion of blood flow was used in order to reduce the computational expense for preliminary insights with 12 kA/m (peak) AMF and an MNP concentration of 5 mg Fe/cm^3^ in the tumor. Maximum temperatures of 40 °C and 44 °C in the liver and the tumor, respectively, were predicted. Overall, these results reliably demonstrated the potential heating efficacy of MHT within reasonable safety limits of applied AMF, assuming (non-physiological) constant perfusion. Similarly, using T1-weighted MR imaging, Fuentes et al. experimentally validated computationally predicted thermal damage boundaries from RF (probe) ablation in bovine livers [67]. Artificial ex vivo perfusion (3.6–53.6 kg/s/m^3^) was performed during the RF ablation treatment to validate the influence of perfusion on the treatment outcome. Using Dice similarity coefficients (DSC), they showed a good overlap between the segmented ablated regions from MR images and the predicted boundaries of ablated regions from the simulations (DSC > 0.7).

### 2.8. Perfusion Imaging Is Essential for MHT Treatment Planning to Capture Dynamic Tissue Responses

A challenge complicating modeling efforts is dynamic tissue response to heating. Blood flow facilitates thermal regulation by dissipating heat from warm regions (e.g., heat sources (MNPs)) to the surrounding tissues and eventually to the environment at the air–skin interface, or through respiration. However, blood flow can increase with modest thermal dose, or it can be significantly reduced if extensive tissue and vascular damage occurs from high thermal doses. These have important implications for therapy outcomes: poorly perfused tumors have been shown to specifically benefit from hyperthermia in combination with radio- or chemotherapy because heat is retained longer, thereby selectively increasing the local thermal dose. Accounting for the thermal effects of tissue-specific blood perfusion is essential in developing accurate MHT treatment plans. Several mathematical constructs have been developed to account for these changes [68].

The blood perfusion term, $\omega_{b}$, is often described as a constant value to minimize the computational complexity. However, a more realistic model will couple both the temperature dependence of the perfusion with a temperature dependent tissue damage term, often assumed to have an Arrhenius relationship to temperature [29]. More detailed temperature-dependent perfusion models capture the non-linear temperature-dependence with as an initial increase in perfusion due to vasodilation during mild hyperthermia and a reduction with a further increase in the temperature beyond mild hyperthermia due to increased vascular stasis. An empirical expression of this non-linear relationship is common [69].
$$\omega_{b}=\omega_{0}\left(30\cdot DS+1\right)\;\;\;\;\;\;\;\;if\;0\leq DS\leq 0.02\;\omega_{b}=\omega_{0}\left(-13\cdot DS+1.86\right)\;\;\;\;\;\;if\;0.02\leq DS\leq 0.08\;\omega_{b}=\omega_{0}\left(-0.79\cdot DS+0.884\right)\;\;\;\;if\;0.08\leq DS\leq 0.97\;\omega_{b}=\omega_{0}\left(-3.87\cdot DS+3.87\right)\;\;\;\;\;if\;0.97\leq DS\leq 1.00$$
(2)DS = vascular stasis = 1−𝛼
where *DS* is vascular stasis; α is the temperature dependent survival fraction; and $\omega_{0}$ is the baseline perfusion at 37 °C, derived from diagnostic imaging as described below.

### 2.9. MHT Treatment Planning with Perfusion Modeling

CT perfusion imaging using a dual-input single compartment model to model the tracer kinetics in the liver has been used to develop heat-based vascular stasis terms [70], where fitting parameters are derived from the enhancement curves of an iodinated contrast tracer in the aorta, portal vein, and liver parenchyma, here:(3)$$\frac{dC_{L}}{dt}=k_{1a}C_{a}\left(t\right)+k_{1p}C_{p}\left(t\right)-k_{2}C_{L}\left(t\right)$$

With tracer concentrations $C_{L}$, $C_{a},\;C_{p}\;$(in Hounsfield units) in the liver, hepatic artery, and the portal vein, respectively, and *k*_1*a*_, *k*_1*p*_, and *k*_2_ are the arterial, portal venous inflow, and liver outflow rate constants, respectively. The fitting parameters can be used to derive basic perfusion parameters such as arterial and portal venous flow and mean transit time, from which heat-based damage can be inferred. Other fitting methods include the single-input, single-compartment model, where arterial and venous contributions are not recognized as separate inputs [71], and deconvolution techniques, where transfer functions defined over a range of flow rates are used to calculate the perfusion parameters [72].

The difference in the thermal damage induced in a tissue with constant vs. temperature-dependent perfusion can be significant in simulations, assuming that constant perfusion usually predicts longer treatment times. Conversely, including temperature-dependent perfusion often produces predictions of significantly shorter treatment times, as vascular damage from heat results in retained thermal energy within the tumor, increasing the damage rates. 

To validate perfusion changes during MHT and simultaneously measure MNP distribution, non-invasive imaging methods such as dynamic, contrast-enhanced magnetic resonance perfusion imaging (DCE-MRI), or dynamic contrast-enhanced computed tomography (CT) perfusion imaging may be used [73,74]. However, artifacts produced by high MNP concentrations in DCE-MRI may preclude clinical translation [73,74]. CT perfusion imaging is unaffected by high intratumor MNP concentrations and may prove more useful to evaluate tumor perfusion changes following MHT. Moreover, advanced volumetric CT scanners that provide a dual-energy option enable concurrent evaluation of tumor perfusion and iron overload, without the presence of distorting artifacts and an almost 4-fold lower radiation dose exposure than perfusion CT [75,76]. Mapping the iron distribution with concurrent tumor perfusion via clinically relevant imaging techniques may prove advantageous for MHT planning, delivery, and monitoring. Presently, cone-beam CT (CBCT) is used to measure the parenchymal blood volume (PBV) parameter instead of dynamic perfusion or flow per se.


Figure 2(**A**) Liver physiology describing the liver acinus comprising the portal triad (portal vein, hepatic artery, and bile duct), sinusoids flanked by hepatocytes and fenesterated endothelial cells, and the draining central vein [54,55]. Figure reprinted with permission from [54] cited in text. (**B**) Based on the distance from the portal triad, the acinus is divided into three zones that are characterized by different permeabilities and metabolic functions in the hepatocytes due to the difference in oxygenation level [56] (**C**). 2D-finite element modeling of liver perfusion based on the liver acinus structure and fluid flow and mass transfer based on porous media theory. PV—portal vein, HA—hepatic artery, BD—bile duct, TPV—terminal portal venule, THA—terminal hepatic arteriole. 1, 2, and 3 indicate different zones in the acinus. The longer edge in the finite element grid represents the TPV, and the shorter edge represents a THV [74]. Figure reprinted with permission from [74] cited in text.
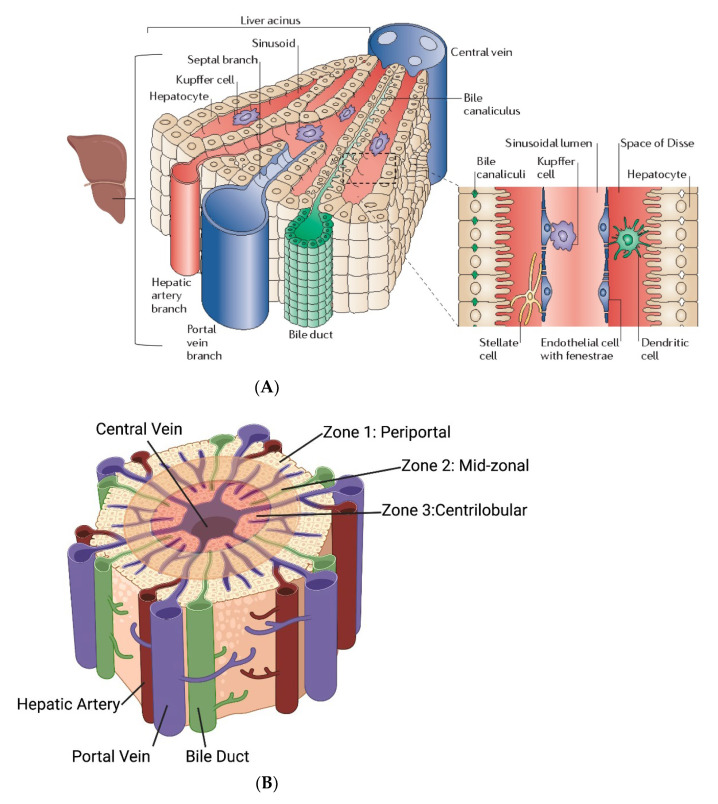

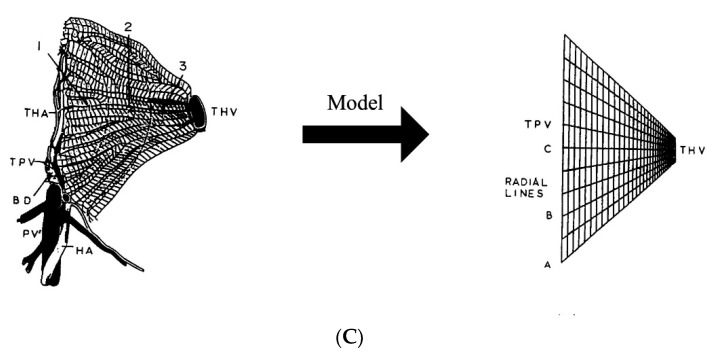



### 2.10. Need for Computational Modeling at the Liver Microcirculatory Level for MHT Treatment Planning

Clinically, hyperthermia is used as an adjunctive therapy in combination with chemotherapy and/or RT. While the input/output tracer kinetic models described previously may provide sufficient information on the perfusion parameters for diagnostic imaging based on step responses to bolus injections of contrast tracer, these models fail to capture the complex fluid and mass transfer processes during therapy (e.g., MHT + TACE). The irregular geometry of the liver acinus and interconnections among sinusoids results in a more complex fluid flow and mass transfer. Additionally, alteration of the acini zones from neoarterial non-triadal branches and portal venous-system shunting, resulting from the tumor, further complicate this fluid flow and mass transfer. Thus, image-guided modeling of microcirculatory physiology of the liver acini is needed to describe the spatiotemporal distribution of the chemotherapeutic, which should be coupled with heat transfer resulting from MHT. Lee et al. applied fluid flow and mass transfer principles from porous media models to the liver acinus to determine the local velocity and pressure profiles (Figure 2) [77]. They used the local velocities and pressures with the conservation of mass equations to determine the concentration of the chemical species in the fluid, which they coupled with mass transfer equations across the hepatocyte cell membrane for a given species. Assuming average physiological parameters such as porosity and hydraulic permeability as a function of radial distance from the central vein (accounting for different zones), they used a 2D-finite element method to simulate the fluid flow and mass transfer. The sensitivity of the model to hydraulic permeability was also shown. To simulate clinical MHT in combination therapies, comprehensive models such as those described by Lee et al. could be coupled with heat transfer models to determine the target and off-target heating. Additionally, HT-chemo interaction terms may be defined using empirical data to reliably predict the therapeutic outcome [78]. The validation of mass transfer models can be performed with the ex vivo normothermic perfusion of livers using venous effluent data and functional optical imaging [77,79,80].

### 2.11. MHT Improved by Amplitude and Power Modulation

As with all thermal therapy techniques, a principal challenge for MHT is to deposit a therapeutic thermal dose within the target, while sparing the surrounding healthy tissue. Thus, the control of energy deposition is important. Depending on the details of their magnetic properties, MNPs can generate significant heat through hysteresis losses when exposed to low RF (~100 kHz to ~1 MHz) alternating magnetic fields (AMF). The heat generated in the tumor (i.e., Q_p_ in Equation (1)), is a function of the heat generated per second per unit mass of the MNPs, defined as the specific loss power (SLP, W/g Fe), the MNP distribution, and their concentration within the tumor: (4)$$Q_{p}\;(x,\;y,\;z)=\sigma \left(x,y,z\right)\times SLP\times c_{Fe}$$
where σ(*x*, *y*, *z*) describes the MNP spatial distribution and *c_Fe_* is the iron concentration. SLP is experimentally determined using calorimetric methods [81,82]. For a given MNP, the SLP generally varies non-linearly with the applied magnetic field (*H*) and linearly with frequency, *f* [29,83]. The nature of the non-linear dependence on the applied field amplitude can be controlled using synthesis methods that modulate the MNP intrinsic variables such as anisotropy, composition, magnetic structure, and size [17,19]. For example, iron oxide nanoparticles comprising aggregated multi-crystallite cores can display overall higher anisotropy than single-domain single-crystallite MNPs, and therefore can generate higher heating rates (from larger hysteresis areas) [17,18]. Similarly, magnetic nanowires can be synthesized by having large anisotropies that can be tuned as a function of their shape aspect ratios, generating large hysteresis areas under AMF [84,85]. However, in these cases, large magnetic field amplitudes (greater than the anisotropy field of the MNP, *H* > *H_k_*) are needed to exploit the larger hysteresis area, which may exceed the clinical limits. An important patient safety constraint defining the limits of patient exposure to AMFs for MHT is the non-specific Joule heating of tissue resulting from induced (Foucault or circular) eddy currents, which depends on the tissue electrical conductivity and scales as (*H* × *f* × *r*)^2^, where *r* is the radius of the eddy current path, often having dimensions related to the dimensions of exposed tissue. Consensus opinion of the limit of exposure of a human torso, *r* ≅ 15 cm is *H* × *f* < 4.85 × 10^8^ A/(m⋅s) [86]. These limits impose constraints on the MNP magnetic properties to produce effective SLP (>200 W/g Fe) @ *H* < 12–14 kA/m and 150 kHz. Methods to obtain large heating rates under low magnetic fields by lowering the anisotropy barrier in multi-crystallite MNPs using processes such as hydrothermal aging have shown promise [17]. Probing the magnetic structure of such MNPs using small angle neutron scattering with polarized neutrons has provided insights into the domain structure, motivating the rational design of MNPs for high SLP [17,18]. 

The spatial control of deposited energy has been challenging, especially due to the heterogenous deposition of MNP heat sources within the tumor, which can be attributed to variables that include tumor heterogeneity and variable injection parameters [87,88]. Constant AMF in such cases can produce a combination of ablated and under-treated zones within the target while ablating regions of healthy tissue. Various amplitude (*H*) and power modulation schemes have been proposed to exploit the non-linear dependence on the field amplitude of the MNP heating to manage energy deposition. For example, Soetaert et al. showed that *H*-amplitude modulation with a low-frequency square wave (2 Hz) could enhance selective energy control to spare healthy tissue when compared to an unmodulated AMF [29] (Figure 3). Kandala et al. showed that amplitude modulation with proportional-integral-derivative (PID) control of temperature with temperature feedback at the tumor–tissue boundary in simple 2D- and 3D-geometries approximating a liver cancer model resulted in better thermal dose distribution (higher CEM43) in the tumor, with enhanced specificity for various (theoretical) homogeneous and (image-based) heterogeneous MNP distributions [30] (Figure 3B). Experimentally, only a few setups for the PID control of temperature during laser ablation and focused ultrasound-based therapies have been demonstrated in in vitro, ex vivo, or in artificial xenograft models [89,90,91]. Power modulation methods have been tested in preclinical models to minimize Joule heating by leveraging the thermoregulatory response (metabolic, physiological, and molecular effects) in tissues [92,93]. However, large clinical scale magnetic hyperthermia systems with temperature feedback control and automated power modulation capabilities remain unavailable.

### 2.12. Theranostic Nanoparticle HT Requires Integrated Imaging

An MNP formulation approved for both diagnostic imaging and MHT is currently unavailable. The required performance specifications for diagnostic imaging with MRI are typically incompatible with those for optimal MHT [19,31]. For MRI, ideal MNPs possess large and reversible magnetic moments (i.e., they exhibit “unblocked” behavior or “superparamagnetism” (SPM) [33]). Paramagnets display a zero area hysteresis loop, and thus exhibit no heating. Accordingly, many MRI formulations are unsuitable for MHT because they generate negligible heat. Furthermore, an integrated MRI/MHT device is physically unattainable because the large static magnetic field integral to MRI effectively prevents the moment traversals are required for dissipative loss power heat generation from MNPs. There is no clinical device approved for combined imaging and therapy with MNPs. This limits the evaluations of MNP intra-tumor distribution and concentration to post-mortem analysis of tissue samples [23]. A new technology is needed that integrates high-resolution MNP imaging with MHT for image-guided therapy. The physical principles enabling magnetic particle imaging (MPI) are compatible with MHT, with the added enhancement that MPI-like technology offers a natural method to spatially confine the region of heating [31]. Thus, MPI technology offers a path for imaging-integrated MHT.

### 2.13. In Vivo Testing of MHT

While simulations can assist with planning MHT treatment, in vivo testing is required for validation before translation to the patients. Several murine models of HCC are available, however, modeling the selective delivery of MNPs to liver tumors via angiographic means is technically challenging in mice due to their small size. Furthermore, mouse liver physiology and vascular structures differ substantially from those in humans. Experimental large animal models such as the rabbit VX2 tumor model are used extensively as a technical model to validate imaging-guided delivery techniques. VX2 is a rabbit squamous cell carcinoma, which is initially grown in the skeletal muscle of a donor rabbit, and then transplanted to the liver of a recipient (naïve) rabbit. The tumor grows rapidly in the liver with a characteristic hypervascular capsule [94]. As the rabbit vascular anatomy [95,96] is similar to the human vascular anatomy, the rabbit is often used for interventional angiographic techniques with the added benefit that conventional human-sized imaging systems can be utilized. The biggest drawback to the rabbit VX2 model is that tumor implantation is performed in a normal rabbit liver having no underlying disease. In addition, the rapid tumor growth often leads to a necrotic core with tumor kinetics that poorly reflect HCC in humans. Similarly, other chemically-induced models of HCC (e.g., carbon tetrachloride) may have an inflammatory component that leads to inconsistent degrees of fibrosis vs. HCC [97]. 

Recently, a genetically altered porcine model, oncopig cancer model (OCM), has provided a translational large animal model of HCC [98]. Cre recombinase exposure, typically through the maintenance field in this genetically manipulated pig, results in HCC development, often with fibrosis that recapitulates human HCC histologically as well as phenotypically, with response to imaging characteristics and response to chemotherapeutics that are similar to human HCC [99]. This model will likely gain in popularity if it becomes more readily available and less expensive.

Interestingly, the woodchuck (*Marmota monax*) develops HCC naturally from neonatal exposure to the woodchuck hepatitis virus [100], and this model has enabled the development of intra-arterial therapies in a larger animal with many similarities to virally induced HCC [101,102,103] in humans. The disadvantage of this model is the long time (>1 y) needed for the development of spontaneous HCC and the relative intractability of the handling of this animal for research purposes. Another spontaneous model of HCC occurs in companion dogs. While liver cancer is relatively rare in dogs, HCC is the most common type, representing >50% of liver neoplasia in dogs with histopathological similarities to human HCC [104]. While probably not the first-line for testing MHT, companion dogs with HCC present a superior model to develop plans for human clinical trials as they are often exposed to many of the same environmental factors as humans, are immunocompetent, have greater genetic diversity than genetically manipulated models, and have owners who are willing to pursue advanced therapies and imaging for their pets. Thus, dogs may be uniquely well-suited for showing the efficacy of MHT in HCC in a large population that can ultimately benefit dogs as well as people. 

Finally, there is increasing interest to use MNPs as drug delivery vehicles that deliver chemotherapy with MHT [105]. This combination is purported to offer enhanced effectiveness with improved distribution in the tumor, and can overcome multidrug resistance in chemo-resistant cancers such as HCC [42,75,106]. In many respects, this approach is similar to ThermoDox^^®^^, a thermosensitive liposomal formulation that releases a payload of doxorubicin when heated by an external energy source (e.g., radiofrequency) for the treatment of advanced HCC [20,107,108]. ThermoDox^^®^^ has undergone testing in clinical trials and is deemed safe [109]; however, the analysis of evidence suggests that to achieve efficacy superior to other modalities, measured by progression-free survival and overall survival, the thermal “dwell time” must be ≥45 min. ThermoDox^^®^^ is delivered i.v., which is important as this provides critical data to inform the viability of systemic delivery approaches with drug eluting nanoparticulate formulations. The ultimate regulatory fate of ThermoDox^^®^^ will undoubtedly influence the fate of other systemic thermal-based drug delivery nanoparticle formulations. Presently, there is no indication that systemic delivery of MNPs carrying drug(s) for HCC treatment will be more effective than local imaging-guided interventional delivery strategies, or that they will receive more favorable regulatory review. 

## 3. Conclusions and Future Directions

MHT is a powerful non-invasive technology that can provide patient-specific cancer therapy. MHT with magnetic nanoparticles offers advantages over other theranostic technologies including those based on optical modalities such as NIR-imaging and photothermal therapy. Tissue is diamagnetic and does not attenuate magnetic fields. Thus, magnetic detection offers quantitative imaging remotely, even for deep-seated tissues, regardless of the tissue type (e.g., through bone, fat, muscle, etc.). Taken together, MNPs paired with the appropriate device enable a path toward integrating imaging, thermometry, and therapy into a single and an unprecedented diagnostic, treatment planning, treatment execution with real-time monitoring, and treatment follow-up platform. When used in combination with other standard of care therapies, MHT has demonstrated survival benefits. However, its full clinical potential for treating HCC has yet to be realized with further technological improvements. 

## Figures and Tables

**Figure 1 nanomaterials-12-02768-f001:**
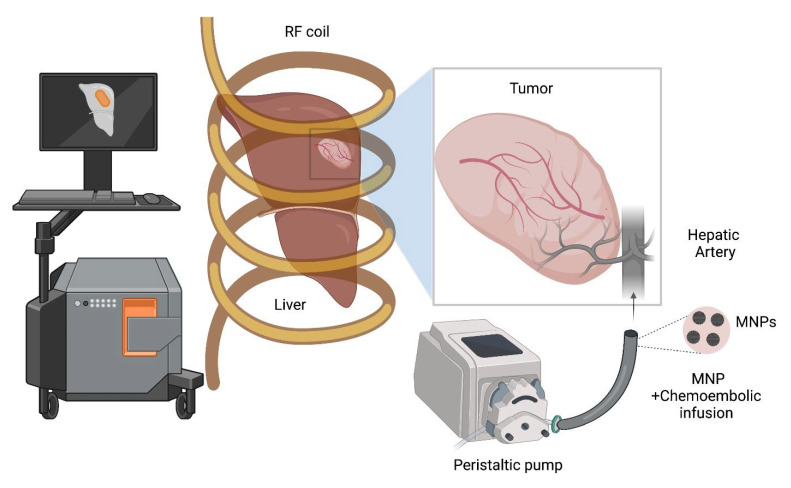
A schematic depicting the image-guided magnetic hyperthermia therapy (MHT) combined with TACE in hepatocellular carcinoma (HCC). Imaging modalities with high contrast and spatial resolution (e.g., MRI for contrast resolution, CT for spatial resolution, and MPI for both) allow for the characterization of MNP distribution relative to the liver lobes, tumor, and vasculature, following intra-arterial perfusion of MNPs with chemo-embolics. This enables patient-specific treatment planning and implementation of the MHT treatment (e.g., temperature probe placement, power and amplitude modulation schemes).

**Figure 3 nanomaterials-12-02768-f003:**
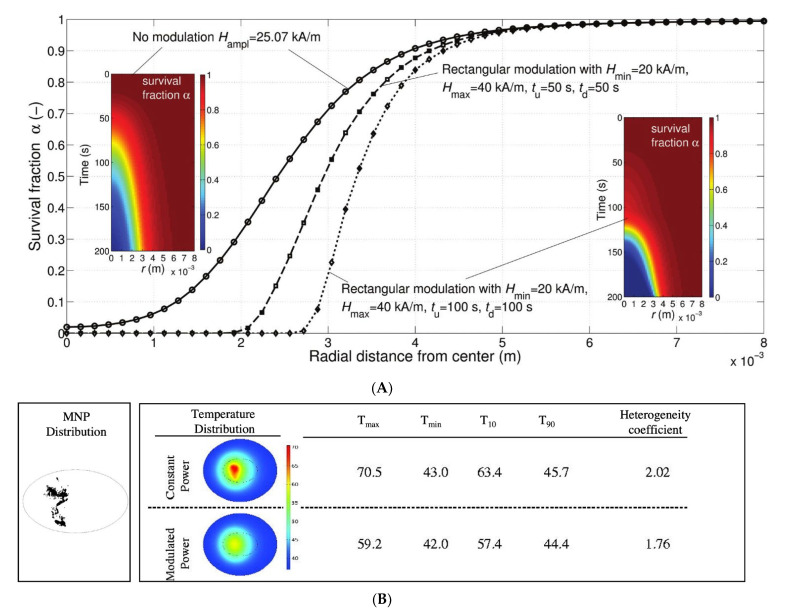
(**A**) Survival fraction (α) as a function of radial distance in a 1.24 cm diameter spherical liver tumor and the surrounding parenchyma when AMF is subjected to a rectangular amplitude modulation vs. no modulation. Rectangular modulation results in greater selectivity of the tumor vs. surrounding tissue with regard to thermal damage. The rectangular modulation is assumed to vary between constant AMF minimum *H_min_* during time *t_d_* and constant AMF maximum *H_max_* during time *t_u_*. Figure reprinted with permission from [29] cited in text. (**B**) The temperature distributions achieved in liver tumor and healthy tissue after 20 min of heating by constant power vs. power modulation with PID control using temperature feedback at the tumor–healthy tissue boundary in the image-derived MNP distributions. Figure reprinted with permission from [30] cited in text.

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
