# Peer review of "Current Challenges in Image-Guided Magnetic Hyperthermia Therapy for Liver Cancer"

_nanomaterials, 2022, doi:10.3390/nano12162768_

Round 1
Reviewer 1 Report
Authors present an interesting review paper of magnetic hyperthermia focused on liver cancer. The review is clear and points out the main limitations to locally control the local heating in a highly vascularized organ, such as liver.
I just have two minor, comments:
1) Maybe the section related to percussion imaging is too long, and could be more clearly focused on the problematic of the magnetic hyperthermia treatment.
2) The quality of figure 2 is poor and the letter size is unreadable.
3) The variables in the equation of line 269, are not defined.
Author Response
We thank the reviewer for the feedback to improve our manuscript. We have revised the manuscript, and below we provide point-by-point responses. Changes in the manuscript are highlighted for easy reference.
Authors present an interesting review paper of magnetic hyperthermia focused on liver cancer. The review is clear and points out the main limitations to locally control the local heating in a highly vascularized organ, such as liver.
I just have two minor, comments:
1) Maybe the section related to percussion imaging is too long, and could be more clearly focused on the problematic of the magnetic hyperthermia treatment.
We thank the reviewer and agree. We have removed section 2.7 and revised and moved 2.8 (now 2.9) to appropriately reflect the utility of perfusion imaging in MHT treatment planning.
2) The quality of figure 2 is poor and the letter size is unreadable.
Figure 2 has been revised to improve clarity and resolution in compliance with journal guidelines.
3) The variables in the equation of line 269, are not defined.
We thank the reviewer for pointing out our oversight. All variables in all equations are now defined.
Reviewer 2 Report
Sharma et al. has presented a well written summary on magnetic nanoparticles and its image guided therapeutic application on hyperthermia. I really appreciate their effort. However, there are some comments I would like to suggest to the authors which would improvise the review article.
1. The authors should improve the quality of figure 3. The figure labels are difficult to read and the table on the right side has texts which have disappeared such as heterogenicity … (the rest cannot be read as it not visible.
2. In figure 2, the internal labels are not clear. The authors should improve the figure presentation.
3. The introduction is quite short. The authors should elaborate the introduction section and discuss how and what are the advantages of MRI imagining of tumors compared to nanosystems with targeted NIR or other dye based imaging with literatures such as https://doi.org/10.1016/j.ejpb.2020.05.002 ; https://doi.org/10.1021/acsami.1c21655
4. How is MRI hyperthermia better than Phototheranostic nanoparticles? The authors could discuss this in the section 2.2. or in the conclusion.
5. In the section 2.2 authors should discuss some literatures where magnetic nanoparticles were not just for imaging but for the therapeutic application as well such as https://doi.org/10.1016/j.carbpol.2011.11.033; https://doi.org/10.1016/j.addr.2008.03.018
6. What are the prospects of MNPs hyperthermic for target specific delivery such as using cancer specific ligands?
Author Response
We thank the reviewer for the feedback to improve our manuscript. We have revised the manuscript, and below we provide point-by-point responses. Changes in the manuscript are highlighted for easy reference.
Sharma et al. has presented a well written summary on magnetic nanoparticles and its image guided therapeutic application on hyperthermia. I really appreciate their effort. However, there are some comments I would like to suggest to the authors which would improvise the review article.
- 1. The authors should improve the quality of figure 3. The figure labels are difficult to read and the table on the right side has texts which have disappeared such as heterogenicity … (the rest cannot be read as it not visible.
Figure 3 has been corrected, revised, and improved in quality and resolution for better readability.
- In figure 2, the internal labels are not clear. The authors should improve the figure presentation.
Figure 2 has been revised for better clarity and presentation.
- The introduction is quite short. The authors should elaborate the introduction section and discuss how and what are the advantages of MRI imagining of tumors compared to nanosystems with targeted NIR or other dye based imaging with literatures such as https://doi.org/10.1016/j.ejpb.2020.05.002 ; https://doi.org/10.1021/acsami.1c21655
We have included a new paragraph and the suggested references.
- How is MRI hyperthermia better than Phototheranostic nanoparticles? The authors could discuss this in the section 2.2. or in the conclusion.
We have followed the reviewer’s suggestion and added text in the Introduction and Conclusion sections.
- In the section 2.2 authors should discuss some literatures where magnetic nanoparticles were not just for imaging but for the therapeutic application as well such as https://doi.org/10.1016/j.carbpol.2011.11.033; https://doi.org/10.1016/j.addr.2008.03.018
We have added a paragraph in section 2.13 (page 11) along with references suggested.
- What are the prospects of MNPs hyperthermic for target specific delivery such as using cancer specific ligands?
We have added a paragraph in section 2.13 and added citations to references.